# RNA Interference in Insects: From a Natural Mechanism of Gene Expression Regulation to a Biotechnological Crop Protection Promise

**DOI:** 10.3390/biology13030137

**Published:** 2024-02-21

**Authors:** Beltrán Ortolá, José-Antonio Daròs

**Affiliations:** Instituto de Biología Molecular y Celular de Plantas, Consejo Superior de Investigaciones Científicas-Universitat Politècnica de València, 46022 Valencia, Spain

**Keywords:** RNAi interference, insect pest, small RNA, double-stranded RNA, recombinant RNA, RNA delivery

## Abstract

**Simple Summary:**

Large amounts of conventional insecticides are used to control insect pests in agricultural production worldwide. New sustainable strategies, more specific and environmentally friendly, are urgently needed to combat insect pests. RNA interference (RNAi) is a natural mechanism of gene expression regulation that has been repurposed for biotechnological applications. Double-stranded RNAs homologous to endogenous genes, which could be produced in the crop plant or exogenously synthesized and applied onto crops, efficiently trigger gene silencing in insects, reducing pest damage. Consequently, these molecules are envisioned as the new generation of insecticidal compounds.

**Abstract:**

Insect pests rank among the major limiting factors in agricultural production worldwide. In addition to direct effect on crops, some phytophagous insects are efficient vectors for plant disease transmission. Large amounts of conventional insecticides are required to secure food production worldwide, with a high impact on the economy and environment, particularly when beneficial insects are also affected by chemicals that frequently lack the desired specificity. RNA interference (RNAi) is a natural mechanism gene expression regulation and protection against exogenous and endogenous genetic elements present in most eukaryotes, including insects. Molecules of double-stranded RNA (dsRNA) or highly structured RNA are the substrates of cellular enzymes to produce several types of small RNAs (sRNAs), which play a crucial role in targeting sequences for transcriptional or post-transcriptional gene silencing. The relatively simple rules that underlie RNAi regulation, mainly based in Watson–Crick complementarity, have facilitated biotechnological applications based on these cellular mechanisms. This includes the promise of using engineered dsRNA molecules, either endogenously produced in crop plants or exogenously synthesized and applied onto crops, as a new generation of highly specific, sustainable, and environmentally friendly insecticides. Fueled on this expectation, this article reviews current knowledge about the RNAi pathways in insects, and some other applied questions such as production and delivery of recombinant RNA, which are critical to establish RNAi as a reliable technology for insect control in crop plants.

## 1. Introduction

In a current scenario characterized by the need of increasing plant production to meet food needs of a world population in continuous growth and the rise of concerns about the environmental impact of human activity, the development of innovative solutions is required to optimize the crop yield, with improved nutritional properties and resistance to all kinds of stresses. In this sense, insect pests destroy around 20% of the worldwide annual agricultural production, with an estimated cost of around 470.000 million dollars [1,2], considering both the productive losses and the increase in costs due to pest control systems. In addition, most plant viruses are transmitted by insects and are benefiting from the emergence of new pests to increase their host range and geographic distribution, thus reinforcing the need of vector-resistant plants to reduce viral diseases [3,4,5]. In the same way, insects act as vectors of important human and animal diseases (with consequential risk of zoonoses), such as malaria, dengue or chikungunya disease [6]. Integrated pest management programs (IPMs) are currently being implemented, which along with good agricultural practices and pest monitoring, combine various control strategies such as baited traps with sexual pheromones or male lures, more eco-friendly new generation pesticides or releasing of sterile insects and pest predators, parasitoids and pathogens [7]. The development of genetically edited plants capable of providing protection against insects, by modifying mixtures of volatiles or expressing *Bacillus thuringiensis* entomotoxins against specific pests, is also promising [8]. In this sense, an alternative that is arousing great interest is the exploitation of the insects’ natural mechanism of RNA interference (RNAi) for their control.

RNAi describes a series of mechanisms highly conserved in eukaryotes, regulating gene expression and protecting against exogenous and endogenous genetic elements, such as viruses, viroids, or transposons. RNAi is triggered by the presence in the cell of small RNAs (sRNA) with high sequence homology to the genetic element to be regulated or protected from. Silencing can occur at the transcriptional and post-transcriptional level; the first involves epigenetic modifications in DNA and histones that repress the transcription process, and the second, mRNA degradation or translational repression.

## 2. RNAi Discovery

The discovery of the RNAi mechanistic bases is attributed to the work of Andrew Z. Fire and Craig C. Mello with the nematode and model organism *Caenorhabditis elegans* [9], which earned them the 2006 Nobel Prize in Physiology or Medicine. Their work established double-stranded RNA (dsRNA) as the main effector of RNAi silencing. They called this phenomenon “RNA interference” to distinguish it from previous gene silencing techniques using antisense RNAs [10]. Their discovery served to explain previous phenomena of unexpected silencing in various organisms. For example, in the nematode, similar levels of silencing were achieved by using antisense and sense RNAs (the latter frequently used as control in antisense strategies) [11,12]. Other examples were described in plants and fungi, in which the use of transgenes to overexpress endogenous or exogenous proteins (with high sequence homology) sometimes resulted in the silencing of these genes [13,14,15,16,17]. All these experimental procedures led to the cellular accumulation of dsRNAs, and consequently, the activation of the RNAi machinery. This seminal work also started a rapid race to identify this mechanism in other organisms, confirming its presence in other eukaryotes, such as plants and animals (including insects and mammals) but not in *Saccharomyces cerevisiae* [18,19,20,21], as well as to unveil the basic cellular components that mediated the dsRNA-induced gene silencing. Ever since, RNAi has become a widely used tool in basic biological research and inspired many biotechnological applications.

## 3. Three Different Pathways of RNA-Mediated Silencing

Several types of sRNA are capable of triggering RNAi responses, which follow different processing pathways and silencing strategies, although the proteins involved are closely related. They include small interfering RNAs (siRNAs), microRNAs (miRNAs), and P-element induced wimpy testis (PIWI)-interacting RNAs (piRNAs). sRNAs can be exogenous (foreign genetic elements, such as viruses with dsRNA genomes, dsRNA replication intermediates or highly structured RNAs, or experimentally introduced RNAs) and endogenous (genome-encoded and transcribed in the nucleus). An additional process regulating gene expression has also been described, known as RNA activation (RNAa), in which the machinery involved in RNAi has evolved to positively regulate the expression of target sequences at the transcriptional level in many eukaryotes [22,23].

### 3.1. siRNAs

The first major pathway, mediated by siRNAs (Figure 1), is triggered by the presence of perfectly complementary dsRNAs or RNA hairpins in the cell cytoplasm. It is mainly involved in silencing exogenous RNAs [24,25], although multiple subtypes of siRNAs with endogenous origin (esiRNA) exist. esiRNAs can derive from transposable elements, genomic regions with inverted repeats and from mRNAs with overlapping sequences (natural antisense siRNA, natsiRNA) [26,27,28,29,30]. esiRNAs are involved in maintaining genome stability [26,27,28] and regulation of gene expression in certain cellular processes, such as energy homeostasis or the response to several stresses [31,32,33].

dsRNAs are recognized and processed by multidomain enzymes belonging to Dicer and Dicer-like (DCL) family of proteins, dsRNA-specific bidentate endoribonucleases that generate shorter double-stranded molecules called siRNA [34,35,36,37]. In *Drosophila melanogaster* and presumably all insects, there are two different Dicer genes, Dicer-2 being responsible for specifically producing siRNAs [38]. *Drosophila* Dicer-2 shares a common architecture with human Dicer, with six general domains: N-terminal helicase, central atypical dsRNA-binding domain (dsRBD), Platform/PAZ (Piwi, Argonaute, Zwille), two tandem RNase III (RNase IIIa and IIIb) and C-terminal dsRBD [39,40]. This enzyme can process dsRNAs according to their characteristics following two different mechanisms [40,41,42,43]. In the first mechanism, the helicase domain recognizes dsRNAs with blunt ends (that are characteristic of viral infection) and consumes ATP to unwind and translocate the dsRNA, placing its termini in the PAZ domain, processively producing siRNA from one end without dissociating the dsRNA. In the second mechanism, the PAZ/Platform domain interacts directly with dsRNAs containing 2 nt 3′ overhangs (that are characteristic of cleavage by RNase III) independently of ATP. After each cleavage, the dsRNA dissociates, and its protruding ends can be recognized again. Both mechanisms are typical, but not exclusive, for the described molecules [44]. The recognition and cleavage of dsRNAs may also depend on the involvement of dsRNA binding proteins (dsRBPs). For example, it has been described that R2D2 (protein with two dsRBDs associated with Dicer-2) prevents Dicer-2 from processing miRNA precursors *in vitro*, increasing its affinity for long dsRNAs [42], while Loquacious-PD (Loqs-PD) is necessary for processing certain esiRNAs but not for viral siRNAs [45,46,47,48]. It has been proposed that Loqs-PD promotes the use of suboptimal dsRNAs by altering the Dicer-2 dependence of terminal structures [43,44,49,50].

The length of the produced RNAs is characteristic of the Dicer protein, since it depends on the distance between the PAZ domain and the active center of the RNase domains, acting as a sort of molecular rule [51,52,53]. Thus, typical insect siRNAs are 20–22 nt long [54]. The presence of two RNase III domains and their characteristic positioning generates 2 nt overhanging 3′ ends, with 5′ phosphate and 3′ hydroxyl termini [42,43,44,45,46,47,48,49,50,55].

Only one strand of the siRNAs (called guide strand) becomes part of the RNA-induced silencing complex (RISC), in which it establishes the specificity of the silencing process by base complementarity with other RNAs [56]; the other strand (passenger strand) is usually degraded [57,58,59]. Strand selection follows the thermodynamic asymmetry rule, whereby the strand with its 5′ termini less stably paired with the complementary strand is preferentially selected [60,61]. The selection occurs in the so-called RISC loading complex, consisting of Dicer-2, R2D2 and possibly additional factors [62,63]. R2D2 interacts with the more thermodynamically stable end of the duplex and tightly binds the 5’ end of the passenger strand, restricting Dicer-2 to the opposite end of the siRNA duplex [64]. This function can be replaced by Loqs-PD in certain esiRNAs [64,65].

The cellular effector of the silencing is a protein of the Argonaute (AGO) family [35,56,66,67], which forms RISC together with accessory proteins and the RNA. In insects, two proteins of the AGO subfamily have been described [66,68,69]. They are multidomain proteins organized in two lobes: the N-terminal contains the variable N- and PAZ domains, connected by a linker (L1); a second linker (L2) connects this lobe to the C-terminal lobe, which contains the middle (MID) and PIWI domains [67,70,71,72]. Both lobes surround a central channel that accommodates RNA. As with Dicer, AGO proteins have specialized functions, with AGO2 being involved in antiviral immunity and regulation mediated by esiRNAs [24,28,30,73,74]. However, the selection of the AGO protein may occur according to the identity of the 5′ terminal nucleotide of the guide strand and the presence and position of mismatches in the sRNA, and not according to the Dicer that generates it. Thus, certain esiRNAs can be loaded in AGO1, and miRNAs passenger strands in AGO2 [75,76,77,78,79,80,81].

The loading of the duplex requires Dicer-2/R2D2 (or Loqs-PD) and AGO2 [58,62,63,82,83]. Their interaction allows the MID domain to recognize and bind the 5′ phosphate end of the guide strand, transferring the duplex [83,84]. Weaker interactions are established between PAZ and the 3′ overhang end [67,85]. The duplex is then unwound, presumably by the N and PAZ domains [63,73,86,87,88], and the PIWI domain cleaves the passenger strand [57,58,59], which is rapidly removed by the C3PO (component 3 promoter of RISC) endonuclease [89]. The 3′ terminal nucleotide is methylated at the 2′-O position by RNA methyltransferase Hen1 [90], possibly to prevent its degradation. A complex of multiple Hsc70-Hsp90 chaperones (70 kDa heat shock analog protein and 90 kDa heat shock protein, respectively) also participates in the transfer [83,91,92], maintaining AGO in an open state throughout the process, at the end of which it hydrolyzes ATP allowing AGO to acquire a closed, mature conformation.

The PIWI domain is responsible for the endonucleolytic cleavage of the target mRNA with base complementarity with the guide RNA. PIWI is similar in structure to RNase H, which typically catalyzes the hydrolytic cleavage of RNA in RNA/DNA duplexes [70,93]. The cleavage generates 5′-phosphate and 3′-hydroxyl ends [94], resulting in an RNA fragment without a poly(A) tail and another without a 5′ 7-methylguanosine cap, which are degraded by exonucleases of the RNA surveillance machinery [95]. The AGO2-mediated cleavage, both in the passenger strand and in target mRNAs, does not occur if there are mismatches between the guide and its complementary strand, partially explaining the differential loading of RNAs in both AGO isoforms [58]. In this case, the elimination of the passenger strand is slower, and silencing is established by blocking protein synthesis.

### 3.2. miRNAs

miRNAs (Figure 2) constitute the most abundant type of sRNAs in animals, regulating multiple biological processes, such as reproduction, development or immunity [96,97,98,99]. They are usually expressed as polycistronic RNAs from intergenic regions of the genome carrying their own promoters or from intragenic regions co-expressed with the gene they regulate [100]. They are generally transcribed by RNA polymerase II as primary transcripts (pri-miRNA) with typical mRNA modifications (5′ cap and 3′ poly(A) tail) [101,102,103] and a general hairpin structure with a long complementary double-stranded region flanked by a terminal loop and two unstructured single-stranded segments (ssRNA).

The pri-miRNAs are processed in the nucleus [104] by the type-III ribonuclease Drosha, with the help of Pasha, a dsRBPs with which it forms the Microprocessor complex [105,106,107]. The processing of pri-miRNAs is especially studied with the human counterpart of the Microprocessor, probably occurring similarly in insects [108,109,110,111,112,113]. Two Pasha proteins recognize and bind the terminal loop of the pri-miRNA, and their dsRBDs interact with part of the dsRNA region. Drosha dsRBD binds the other half of the dsRNA, and additional interactions are established with the terminal ssRNA-dsRNA region. Both Drosha and Pasha contacts ensure the binding of suitable substrates, acting together as a molecular rule and accepting RNAs with two ssRNA regions separated by a dsRNA of about 35 bp. Additional RNA motifs may affect processing efficiency [114]. The two RNase III domains of Drosha eliminate the basal single-stranded segments and part of the dsRNA (~11 nt), generating miRNA precursors (pre-miRNAs), hairpins of about 70 nt with one of the characteristic 2 nt protruding 3′ ends of miRNAs. As an alternative to this canonical biogenesis pathway, one subtype of miRNAs is derived from introns; thus, they are known as mirtrons. For their maturation, they depend on the splicing machinery and debranching enzymes but not on the Microprocessor, directly entering the pathway as pre-miRNAs with a hairpin structure [115,116].

The next step in miRNA processing occurs in the cytoplasm [104]. Exportin-5 (Exp-5) mediates pre-miRNAs translocation to the cytoplasm through the nuclear pores in combination with Ran GTPase [117,118]. The efficient transport of precursors by Exp-5 may depend largely on the presence of 2 nt 3′ overhangs and, to a lesser extent, on the characteristics of the apical loop, as occurs in mammals [119,120,121]. Release of pre-miRNA into the cytosol requires hydrolysis of GTP by Ran GTPase. There, they are processed by Dicer-1 [38,75]. The PAZ domain recognizes and binds to the 3′ overhang generated by Drosha while the helicase domain binds to the loop region [122], the latter being not functional in this isoform [38]. The helicase senses the loop size while PAZ act as a molecular ruler measuring the distance from the loop to the 3′ overhang. Thus, tandem RNase III domains only cleave substrates with adequate distance between the 3′ overhang and the terminal loop [122]. The cleavage removes the loop and generates the second 3′ overhang, resulting in ~22 nt RNA. The PB isoform (and to a lesser extent PA) of Loqs is involved in the processing of most but not all miRNAs, possibly stabilizing RNAs with unstable structures at the cleavage site [123,124,125,126,127].

Next, the miRNA duplex is loaded onto RISC containing AGO1 [73], and the strand selection occurs following the same rules as for siRNAs. Unlike siRNas, in which R2D2 or Loqs-PD are required for AGO2 loading, Loqs-PB seems to be dispensable for the duplex transfer to AGO1 [127], although it is unknown whether Dicer-1 by itself is sensitive to the asymmetry or requires additional factors. The loading is dependent on ATP, possibly to keep AGO1 in an open state. The duplex unwinds in a similar way as in AGO2, and the passenger strand is removed without endonucleolytic cleavage, facilitated by miRNAs’ characteristic mismatches [88,128]. It is worth mentioning that miRNAs and their non-mature forms can be edited, adding or modifying nucleotides that are critical for their maturation, regulation and functionality [129,130].

In animals, the miRNAs recognition elements (MREs) are usually found in the 3′ untranslated region (3′ UTR) of mRNAs [131]. Their interaction is usually imperfect but with a characteristic pattern: nucleotides 2–8 of the miRNA 5′ end (called seed region) have perfect complementarity with the mRNA and is sufficient for its function, although additional pairing may participate in the process [132,133,134]. This short sequence allows a miRNA to regulate several mRNAs and different miRNAs can act on a single mRNA [135,136]. In *D. melanogaster*, both AGOs have cleavage activity; however, the catalytic rate of AGO1 is limited by the inefficient dissociation of the reaction product [75]. Thus, AGO1 induces silencing in several ways in which GW182 proteins generally participate by recruiting cell factors and serving as scaffolding. It can inhibit translation by preventing the interactions between the poly(A) tail and the 5′ cap that pseudocircularize mRNA during translation, as well as preventing ribosomal recruitment by dissociating the eukaryotic initiation factor eIF4F from the cap [137,138,139]. It can recruit deadenylases and decapping enzymes [137,140,141,142,143], making the mRNA sensitive to the action of the exoribonuclease 5′-3′ XRN1 [144].

### 3.3. piRNAs

The piRNA pathway (Figure 3) is mediated by proteins of the AGO PIWI subfamily (Piwi; Aubergine, Aub; and AGO3). It occurs mainly in the germ line and is associated with RNAs originating from genomic repetitive intergenic regions and transposons (piRNA clusters). Its main function is to maintain genomic integrity [145,146]. piRNAs also participate, in germ and somatic cells, in processes such as fertility, maintenance and differentiation of stem cells or defense against some viruses [147,148,149,150,151].

Unlike siRNAs and miRNAs, piRNAs are processed from single-stranded precursors that are transcribed by RNA pol II and processed and exported as typical mRNAs [152,153]. In germline, the expression of precursors from bidirectional clusters is common [146,154,155], involving the same polymerase but requiring specialized cell factors. Rhino recognizes these clusters, generally with characteristic modified histones (H3K9me3), and uses Deadlock as a scaffold to recruit the Moonshiner transcription factor, Cutoff (which prevents transcript splicing and modification) and the mRNA nuclear export factor complexNxf3-Nxt1, that mediates export of precursors to the cytoplasm [155,156,157,158,159,160,161].

piRNA precursors accumulate together with proteins that intervene in their biogenesis in perinuclear and perimitochondrial electron-dense regions in the germline but only in perimitochondrial regions in somatic cells [162,163,164]. In germline cells, the 5′ phosphate end of piRNAs is generated by Aub proteins containing antisense guide RNAs that recognize and cleave sense precursors, or by the Zucchini (Zuc) endonuclease, possibly with the help of additional factors such as Armitage, which recognizes conserved motifs with a moderate preference to those that generate uridines at the 5′ end [165,166,167,168,169]. Only the second pathway occurs in the somatic line, since the machinery required for generating complementary piRNAs is not expressed. The 5′-phosphate intermediates generated in both pathways are loaded onto Piwi proteins (somatic cells) and also in AGO3 (in germline) [146]. Mature piRNAs are slightly longer than miRNAs and siRNAs (23–32 nt). The 3′ end of piRNAs is generated by additional downstream action of Zuc or Aub proteins. In some species such as silkworm, the 3′ ends are trimmed by a 3′-5′ exonuclease to generate optimal RNAs to be fully accommodated by PIWI subfamily proteins [170,171]. In *D. melanogaster*, it has been suggested that the Nibbler exonuclease is dispensable if the 3’ cleavage is generated by Zuc, which would directly generate the appropriate end of the piRNA [172]. The maturation of the piRNAs concludes with the 2′-O-methylation of its 3′ end by the Hen1 methyltransferase [90,173], probably being required for its correct interaction with the PAZ domain, as in mammals [174].

In germline, sense precursors loaded to AGO3 serve as a guide for the cleavage of homologous antisense transcripts [146,175]. The new molecule is recognized, loaded and processed in Aub. piRNA-Aub, in turn, recognizes and cuts transcripts from the piRNA cluster itself, generating new sense piRNAs, thus amplifying the silencing signal with the generation of secondary piRNAs in the so-called “ping-pong mechanism”. Due to the cleavage pattern, the molecules loaded in AGO3 have 10 nt homology to the antisense piRNA, and most of them have an adenine in the tenth position. Also, the additional activity of Zuc/Aub/AGO3 on the precursors generates new intermediates that can be processed and loaded by other PIWIs, generating phased piRNAs that increase the sequence diversity of piRNAs [165,167,168,169].

To carry out their silencing activity, Aub and AGO3 remain in the cytoplasm where they mediate post-transcriptional gene silencing by cleaving target RNAs [146,175], repressing translation and promoting mRNA degradation [176,177,178]. Conversely, Piwi develops its function mainly in the nucleus, inducing epigenetic changes [179,180,181] and deadenylating mRNAs in sites of active transcription of transposable elements [182].

## 4. dsRNA Cell Uptake and Systemic Distribution of the Silencing Signal

The mechanisms described in the previous section are known as cell-autonomous RNAi, while non-cell autonomous RNAi encompasses the processes that occur before and after it. These are the uptake of dsRNA from the extracellular medium (environmental RNAi) and the transport of the silencing signal to other cells (systemic RNAi) [183] (Figure 4). Both processes require additional cellular machinery that seems variable between organisms; thus, less information is known about them. The following sections will focus on those aspects that are relevant for the oral delivery of dsRNAs to insects.

Most of the knowledge about non-cell autonomous RNAi comes from *C. elegans*, in which it is mediated by the endocytosis pathways and systemic RNA interference deficiency (SID) membrane proteins. SID-2 is only expressed in the apical membrane of intestinal cells and mediates the endocytic uptake of dsRNA from the intestinal lumen into the cells, not being required for systemic RNAi; SID-1 is a ubiquitously expressed (except in neurons) dsRNA-specific transmembrane channel that mediates endosomal dsRNA release into the cytoplasm during environmental and systemic RNAi [184,185,186,187]. Two endosomal vesicle-associated proteins, SID-5 [188] and SID-3 [189,190], are also required in both dsRNA uptake and systemic distribution by mediating the import and export of vesicles containing such dsRNAs.

Other than *C. elegans*, SID-2 homologs have only been found within the genus *Caenorhabditis*, in species resistant to environmental RNAi [185]. SID-1 homologous genes have been identified in most insects, except for those of the superorder Antliophora (Diptera, Mecoptera and Siphonaptera) [191]. However, these genes may have more homology with tag-130/CHUP-1, a SID-1 paralogue that does not participate in RNAi in *C. elegans* but is involved in cholesterol internalization [192,193]. This protein has been indirectly related to dsRNA movement by influencing the composition of the plasma membrane. The effect on the uptake efficiency of altering the membrane fatty acids composition has been demonstrated as an immunological mechanism to protect insects from subsequent exposures to environmental dsRNA [194]. This is consistent with the elusive role of this protein in insects, since there is no straightforward association between absence, presence and number of Sid-1-like genes with the efficient uptake and systemic distribution of dsRNAs [192,195,196]. Therefore, other mechanisms must facilitate these non-cell autonomous RNAi in insects.

Studies with *D. melanogaster* S2 cells showed the role of clathrin-mediated endocytosis in the uptake of dsRNAs [197,198]. In addition, chemical blockade of pattern recognition receptors disrupts uptake [197], and two scavenger receptors, SR-CI and Eater, have been implicated as the main mediators of the process [198]. A similar role for these receptors and clathrin-mediated endocytosis has been demonstrated in several insect species [193,195,199,200,201,202,203], while Eater/SR-CI have also been involved in the clathrin-independent phagocytosis of dsRNAs encapsulated in bacteria [204]. However, several of these works show that silencing or blocking these receptors does not completely interrupt the uptake. Therefore, it is likely that dsRNAs can be recognized by different receptors that could differ between species. Similarly, the involvement of other clathrin-independent pathways in the process cannot be ruled out, as has been seen to occur with some dsRNA structures [203].

Interestingly, the recognition of naked dsRNAs by the insect uptake machinery is length-dependent. It has been shown to be efficient for long molecules (greater than ~50 bp) but not for short molecules such as siRNAs [197,205,206], which can be a disadvantage in many dsRNA delivery methodologies. Another factor that limits the development of this type of strategy is the variable capacity of RNases in the digestive tract of different insect species to degrade exogenous dsRNAs [207,208,209,210]. Some insects also have extremely alkaline midguts, thus inducing alkaline hydrolysis of dsRNAs. Similarly, efficient endosomal escape of dsRNAs is required for developing the RNAi response. For example, some species have low sensitivity to RNAi, at least in part, due to dsRNA entrapment in endosomes and degradation after fusing with lysosomes [211,212]. Blocking the interaction of late endosomes with lysosomes enhances si- and miRNA-mediated silencing in *D. melanogaster*, while blocking late endosome formation limits silencing [213], thus restricting dsRNA escape between late endosome formation and lysosomal fusion. The activity of the vacuolar H^+^-ATPase has been related to dsRNA cell entry in several species [197,201,214]; however, the mechanism mediating the endosomal escape is not exactly known.

In some insects, dsRNA delivery can result in the generation of a systemic response by short- and long-distance RNAi signal movement. It is likely that the signal is transmitted by direct intercellular contact through membranous protrusions called tunneling nanotubes (TnTs) that allow the connection between cells. Viral infection in *D. melanogaster* cell cultures induces the formation of these structures that transport dsRNA and components of the RNAi machinery [215]. In addition, as occurs in mammals [216] movement of the RNAi signal through tight junctions could be possible, although not yet demonstrated. Over long distances, dsRNA movement appears to occur through hemolymph. In certain insects, hemolymph nucleases efficiently degrade dsRNA [217,218,219], hampering the systemic response and effectiveness of RNAi. dsRNA transport in the hemolymph is mediated by carrier molecules, thus protecting it from degradation. The dsRNA-binding ability of apolipophorins purified from the hemolymph of *Bombyx mori* [220] and *Schistocerca gregaria* (and probably in species of the orders Orthoptera, Blattodea and Diptera) [221] has been demonstrated, strongly suggesting a conserved mechanism in insects. Apolipoforins are the protein components of lipophorin, hemolymphal lipoprotein complexes which function in lipid transport and are also part of the insect antiviral defense. Lipophorins are scavenger receptor ligands, and in ticks, these receptors have also been implicated in systemic RNAi [199]. RNAs are also carried by extracellular vesicles. miRNA-containing vesicles have been identified in *D. melanogaster* cell cultures [222], and their occurrence *in vivo* has been proposed [223]. Furthermore, viral infection in this species generate viral siRNAs that are packed in vesicles that circulate through hemolymph, systemically diffusing the RNAi signal [224]. As for exogenously supplied RNAs, the encapsulation of long dsRNAs and derived siRNAs has been shown in extracellular vesicles of *Tribolium castaneum* and *Leptinotarsa decemlineata* cell cultures [225,226]. In the latter, some of the factors related to endosomal generation and recycling pathways participating in the process were detailed. The full extent of these mechanisms, as well as the possible involvement of additional factors, have yet to be fully resolved.

Additionally, in some insects, the silencing effects are not restricted to the treated insect but also appear in its progeny, even some time after the administration has stopped [227,228,229,230]. This is the so-called parental RNAi, of which most of the mechanistic details of the transfer are unknown.

## 5. Sources of dsRNAs with Insecticidal Effect

The first strategy used to assess pest control by means of RNAi was to develop transgenic plants expressing specific dsRNAs for silencing [231,232] (Figure 5). Since then, countless reports of plant-produced, dsRNAs-mediated gene silencing in insects have been published, with the first commercial variety approved by competent Canadian, USA and Chinese administrations (2016, 2017 and 2021, respectively). This is SMARTSTAX PRO corn (Bayer), that produces dsRNA against the Snf7 gene of *Diabrotica virgifera virgifera* [233]. Its commercialization is scheduled to start soon. However, multiple limitations hold back the development of this kind of technology. On the one hand, generation of genetically modified organisms (GMOs) is laborious, expensive due to the rigid commercialization regulations [234], and currently have scant public acceptance [235]. On the other hand, plant RNAi machinery recognizes the produced dsRNAs, generating siRNAs [236], which can negatively affect their uptake by the insect. A possibility to circumvent this problem consists of expressing dsRNAs in chloroplasts or other compartments lacking RNAi machinery [237,238,239], although their accumulation is very size-sensitive [240,241], and their usefulness with sap feeding Hemiptera is limited [242]. The use of RNAs with structures resistant to the plant RNAi machinery, such as artificial pre-miRNAs, has also been proposed [243].

Other approaches not requiring plant modification have been developed. Plants infected with modified viruses have been widely used for screening potential RNAi target genes [244,245,246,247], and similar viruses may also confer protection to plants against fungi and nematodes [248,249]. The virus acts as a dsRNA factory during its replication in the plant. The wide variety of vectors commonly used to produce molecules of interest with minimal damage to plants makes them an interesting alternative. Similarly, insect-specific viruses have been used to silence endogenous pest genes in functional genetics, and their use for pest control has been proposed [250,251,252,253,254,255]. Replicating engineered viruses can be useful in those cases in which the insect is resistant to environmental and systemic RNAi, given its ability to transfer its genome into the cells, multiplying and establishing systemic infections. Additionally, it would provide another layer of specificity as viruses can be highly host-specific [253,255]. An interaction between both types of viruses is found in the Flock House virus, which replicates in insects and plants [256,257,258]. Although viruses represent an interesting insecticidal strategy, the cross-kingdom status of the silencing suppressors encoded in both virus types must be considered [259,260]. In addition, the environmental release of transgenic viruses can pose biosafety problems. An alternative is the use of virus-like particles (VLPs) synthesized *in vitro* or in modified microorganisms and plants, expressing the dsRNA of interest and viral capsid proteins that self-assemble into virus-like structures enclosing the nucleic acids. VLPs are usually produced in plants to produce recombinant proteins, but also have been used to induce resistance against viruses and insects [261,262]. Although they lack replicative capacity, they confer protection to dsRNA and retain the potential to transfer dsRNAs to the cytoplasm, in addition to some host specificity. Also, they lack silencing suppressors. With the perspective of continuous production in the insect and host specificity, the use of bacteria and fungi able to parasitize or symbiotize pest insects and modified for producing dsRNA has been studied [263,264,265,266,267].

The exogenous application of dsRNAs as a non-transformative alternative for plant protection was pioneered against viral diseases [268], while the use of dsRNAs pulverized as a conventional pesticide was later proposed [269]. The RNA can also be internalized in the plant by roots and petiole or trunk injection [270,271,272,273]. In these strategies, the dsRNA enters the plant, but it is retained in the xylem and apoplast, thus not being processed by the plant RNAi machinery. High-pressure dsRNA spraying has been shown to induce systemic silencing of plant genes and confer resistance to fungi and insects [274,275,276,277], in a process in which dsRNAs have also been detected in the phloem. These strategies require the production of dsRNAs in heterologous systems, as well as their purification and encapsulation. mRNAs and dsRNAs for vaccines are produced by *in vitro* systems as they are more quickly developed and entail fewer concerns regarding their production in microorganisms. Similar systems are unfeasible for intensive use in the field given the enormous quantities required and their high production cost [278,279], which would increase crop production prices. *In vivo* production methods, using microorganisms as a biofactory, are more attractive. The microbiological variety, easy handling, fast growth and heterologous production capacity make them economically viable. The most widely used procedure is the L4440-HT115(DE3) system [280,281] developed for the initial RNAi experiments in *C. elegans*. It is based on the transformation of plasmid L4440, which contains two opposing T7 promoters flanking the cDNA of the gene to be silenced, into a modified *Escherichia coli* bacterial strain that lacks the dsRNA-specific endonuclease RNase III but has the bacteriophage T7 RNA polymerase under the control of the inducible lac operon. Bidirectional transcription results in two complementary ssRNA strands that hybridize; both the whole bacteria or further dsRNA purification are feasible for RNAi strategies. Multiple advances have been made on this type of system in order to increase performance, such as the use of different strains and microorganisms (including their genetic modification), the development of new expression vectors, the improvement of fermentation and extraction methods, etc. [282,283,284,285,286,287,288]. To put these advances in perspective, researchers initially reported the production of 4 μg of dsRNA per ml of culture with the L4440-HT115(DE3) system [268], while other researchers recently achieved larger than 1 mg/mL using a modified strain of *Corynebacterium glutamicum* expressing a high copy number L4440-derivative [286]. As a result of these improvements and the advent of novel systems, the price per gram of dsRNA produced has been reported to fall from over $12,000 in 2008 to about $1 today, and up to half of that in cell-free systems [279]. Our research group has recently developed a system to overproduce RNAs in *E. coli* based on the intrinsic properties of viroids [289,290,291], obligate plant parasites with minimal genomes of non-coding, single-stranded but highly structured circular RNA. Expressing RNAs of interest within the eggplant latent viroid (ELVd) (+) RNA increases their half-life and accumulation in the bacteria due to the circular viroid scaffold, compact and possibly associated with the ligase [289,290,291]. To produce dsRNAs, the incorporation of a self-processing group-I intron cDNA between the inverted repeat is needed to stabilize the expression plasmids, while the intron RNA is efficiently excised from the final chimera, contributing to its compaction. An additional sequence of the intron in a permuted configuration flanking the inverted repeat allows the production of circular dsRNAs without the viroid scaffold [292,293]. This system is also interesting as some viroids such as ELVd are the only known pathogens able to infect the chloroplast. Lacking RNAi machinery, nucleus-expressed but chloroplast-accumulated chimeras could be a potential new strategy for effective pest control. It is also worth highlighting that although the use of perfectly complementary hairpins or dsRNA, both processed by the siRNA pathway, is the most common methodology for insect control and genetic studies, there are some examples of the miRNA and piRNa pathways being exploited for the same purposes [243,294,295,296,297].

Naked dsRNA molecules are prone to degradation by several biotic and abiotic stresses when used as pesticides. Furthermore, in some insects, they have a limited ability to be efficiently uptaken and systemically distributed. Therefore, they are usually formulated in combination with carrier molecules of different natures that increase dsRNA bioavailability in cells. Most of these strategies are based on the advances made in human therapies, and as a common feature, they have cationic surfaces that allow the interaction and encapsulation of the negatively charged phosphate backbone of nucleic acids, as well as the interaction with the negatively charged cell membrane [298]. One common strategy is to encapsulate dsRNA in liposomes, lipid bilayer spherical structures [299,300,301,302,303]. Multiple commercial transfection reagents, with different lipid compositions, may be useful to improve species-dependent recalcitrances; however, they are usually expensive and, in many cases, potentially cytotoxic, and may affect beneficial insect species. Another widespread strategy is the use of the natural polysaccharide chitosan, due to its abundance, low cost, biocompatibility and degradability [303,304,305,306]. Extensive modifications to these natural polymers have been made to improve their stability and cell delivery in pest control and other applications [307,308,309]. Several inorganic nanoparticles can also be used, such as hydroxyapatite, silica, phosphate calcium, carbon allotropes or quantum dots [303,305,310,311,312]. They have low toxicity and a high surface/volume ratio that allows efficient loading of RNAs. Usually, they are functionalized or associated with polymers of synthetic origin, whose chemical variety allow highly versatile particle designs, thus modulating cytotoxicity, dsRNA stability in specific insect environments, cell uptake, etc. [313,314]. An inorganic nanoparticle that does not need functionalization or association with other polymers is layered double hydroxide (LDH), used in plants to provide resistance to virus, fungi and also for pest control applications [315,316,317,318]. There are also two interesting protein alternatives. On the one hand, branched amphipathic peptide capsules (BAPCs), bilayer structures very similar to liposomes but made up of peptides. Protein nanostructures, such as BAPCs, have been described as potentially more biocompatible and biodegradable than synthetic polymers, and more stable than those composed of lipids and polysaccharides [319]. They have been used to enhance the RNAi response in insects such as *T. castaneum*, *Acyrthosiphon pisum* [320], and *Spodoptera frugiperda* [321]. In the latter case, clathrin-mediated endocytosis and macropinocytosis uptake has been described, along with high endosomal escape and an increase in dsRNA transcytosis to hemolymph, improving the systemic response. On the other hand, peptides can be used as uptake mediators. Cell-penetrating peptides (CPPs), both derived from natural proteins or engineered, are rich in basic amino acids that can establish complexes with dsRNA or coat other nanostructures. Their variety allows dsRNA cell internalization in several ways, thus overpassing recalcitrances to any specific entry pathway, as reviewed extensively for human therapy [322]. This strategy has been successfully used in insects, using a CPP fused with a dsRBD to improve the silencing effect in *Anthonomus grandis* [323]. A similar improvement is obtained with the fusion of a dsRBD with agglutinin in *Spodoptera exigua* [324].

Additional strategies propose increasing RNA bioavailability in cells by reducing dsRNA degradation by combining it with nuclease inhibitors such as EDTA or divalent ions [301,325] or chemically modifying the RNA [326], but also improving the endocytosis process by altering the membrane composition with hydrogen peroxide or arachidonic acid [194,327].

## 6. RNAi in Pest Control: Challenges and Future Directions

The current challenges for exploiting RNAi as an insecticidal strategy can be grouped into (i) the variable efficiency of RNAi among pests, and (ii) the cost-effective production of dsRNA.

It is well known that not all insects are equally susceptible to dsRNA, with enormous differences between insects of different orders but also within closely related species. Additionally, variable silencing efficiency in the same insect is commonly obtained depending on development stages, target tissues and/or delivery systems. Although progress has been made in recent years to unravel the molecular determinants of the RNAi efficiency, being a set of highly interrelated factors, they are still not fully understood. Despite several exceptions reported, generally orthopteran, blattodean and coleopteran insects are considered to be very susceptible to exogenous dsRNAs, while in hemipteran species, the RNAi efficiency is highly variable, and lepidopterans and dipterans usually have much lower efficiencies.

For efficient insect control, the appropriate target gene has to be selected. Silencing and mortality have been described by affecting a wide spectrum of gene functions such as energy metabolism, membrane transporters, detoxification, structural proteins, etc. Ideally, gene silencing should lead to the death of the insect in the shortest time and with the minimum dose of dsRNA. Thus, it must encode ubiquitously expressed proteins with a short half-life originating from abundantly transcribed mRNAs with high turnover. An important detail is that under ideal experimental conditions, insect food intake is controlled and restricted to sources with dsRNA, while in the field, the insect will have food sources beyond those that supply the dsRNA, thus hampering the silencing process. In this scenario, the timeframe in which compensatory upregulation of the target gene or paralogs that could functionally supplement said gene would increase, allowing the compensation of the silencing phenotype. The sublethal effects of RNAi on pests [328,329], as well as parental RNAi, may become essential to reduce crop damage in the field; the mechanisms involved in the latter process must be explored. However, it is also possible that sublethal treatments could facilitate the appearance of refractoriness in some insects, protecting them from subsequent exposures. Worryingly, the first case of resistance development after RNAi treatment has been reported [330]. The resistance was located at a single autosomal locus and inherited recessively; however, we are still clueless of how this resistance occurs and how to minimize it. It is also important to mention that additional factors, such as the environmental fate of RNA and its effects, including the repercussions on non-target species, have to be taken into account. Reassuringly, a myriad of studies showed that pest-specific dsRNAs do not seem to have a negative impact on unrelated species. It is expected that the establishment of high-throughput strategies, along with -omics technologies, will provide the tools to select the best target genes, and rational designs of the silencing strategies must ameliorate some of the mentioned problems. For example, it is believed that increasing homologous sequence of the dsRNA (and therefore, the diversity of derived siRNAs) can hinder the development of resistance, while off-target effects can be reduced by selecting less evolutionarily conserved sequences. In addition, the synergistic effects of targeting multiple essential genes may be interesting to increase the likelihood of insect mortality, avoiding functional supplementation and resistance development. We must also not forget the benefits that the combination of RNAi with other control strategies can provide. In any case, an adequate risk assessment is needed.

A determinant of RNAi efficiency is dsRNA degradation, partially explaining the differences between delivery methods, as in some species, effective silencing by injection is correlated with oral insensitivity. This is especially relevant in lepidopterans, hemipterans and dipterans. Lepidopterans have the highest degradation capacity in insects and express specific nucleases not seen in any other order, reducing the efficiency of RNAi in these insects [209], while Hemipterans are the only reported order to degrade dsRNA in saliva [210]. In addition, dsRNase expression can vary between life stages (resulting in the stage-dependent silencing results). Several “RNAi-of-RNAi” studies showed that silencing mRNAs of specific nucleases managed to increase the induced silencing [331,332,333,334]. Thus, co-delivering strategies have been proposed to extend the scope of this experimental strategy to pest control. It would also be interesting to explore new rationally designed RNA nanostructures to increase the half-life of the dsRNA (both in the field and within the insect) [335,336,337,338], along with the previously mentioned use of nanoparticles or formulations.

Deficient dsRNA uptake limits RNAi responses in dipterans and in certain tissues of orthopterans and hymenopterans, while impaired endosomal escape of dsRNA appears to be limiting in lepidopterans. The mechanisms governing these processes, however, are mostly unclear, possibly involving different mechanisms in different species, stages and/or tissues. Thus, further work is needed to uncover the cell factors involved in how to enhance the uptake. RNAi-of-RNAi strategies could be employed to co-silence factors (such as tag-130/CHUP-1) that hinder RNA uptake if multiple rounds of feeding are going to be required for effective silencing; also, as certain factors are known that mediate endosomal-lysosomal fusion [339], their co-silencing could minimize the fusion, thus increasing the time frame of RNA entry into the cytosol. In this regard, small molecules such as chloroquine have been reported to promote the cytosolic translocation of endocytosed nucleotides for human therapeutics *in vitro* [340]. Again, novel dsRNA structures could help in the uptake processes as previously reported [203], along with the use of nanoparticles or formulations.

Systemic distribution of the silencing signal is generally required to achieve phenotypic effects. But unlike nematodes, fungi or plants, the RNAi signal is not amplified as many insects lack endogenous RNA-dependent RNA polymerases (RdRp) [341]. Thus, all silencing derives from the initial introduced RNA, making adequate concentrations of dsRNA essential, which can be difficult for its application as they are usually high. While it has been speculated that RNA is possibly transported packaged in protective structures such as exosomes, again, the mediators of this process are largely unknown. Extensive research is needed to fully understand the cellular pathways involved and how it can be hijacked to our favor with the use of nanocarriers, small molecules or novel strategies to enhance dsRNA propagation. As an alternative for the most recalcitrant insects, it is expected that potent and localized silencing in the dsRNA-capturing cells could reduce the damage caused by insects. For example, disrupting the proper functioning of midgut cells via RNAi could limit insect feeding and even induce death [342]. On a more positive note, virus-infection-derived secondary sRNAs has been found in *D. melanogaster* [224], being generated in hemocytes through viral DNA synthesis by endogenous reverse transcriptase, followed by transcription and dicing, secreted in exosome-like vesicles and conferring systemic protection. Therefore, it would be interesting to study its occurrence in other insects and if this mechanism could be exploited to enhance the silencing signal.

The RNAi pathways described in Section 3 come mostly from studies in model species such as the insect *D. melanogaster*, or the mammal counterparts, but the core enzymes have been identified in an increasing number of species. Interestingly, its genes have suffered duplications and deletions [191], and its basal expression varies between tissue and stage, thus partially explaining RNAi variability [343,344,345,346]. Furthermore, Dicer and AGO of certain insects may be not equally functional due to different evolution of their structures. For example, a recent study identified variability in certain conserved domains and loop regions of the RNAi machinery, especially in Lepidoptera [347]. Even less known are the accessory protein factors involved in siRNA; they may not be as conserved as the core enzymes, and we may still not know factors relevant to the process. All these aspects could explain, for example, the differential base bias in dsRNA processing between species, affecting the RNAi efficiency [348]. Curiously, it has been proposed that in Lepidoptera (and to a lesser extent in Diptera) the prevalence of viral infections has led to evolutionarily replace the siRNA pathway as the prime antiviral strategy in favor of alternative defense mechanisms that cannot be overcome by viral silencing suppressors (thus diminishing expression/function of core RNAi factors and dsRNA uptake components, and increasing nucleases) [349]. In sum, the mechanistic details of one species cannot be directly extrapolated to other insects. Species-specific studies must be conducted to assess the basal expression (and possible upregulation after treatment) of core siRNA enzymes to target (if possible) life stages and tissues with the highest core expression as well as to identify variations in its mode of function to tailor the characteristics of the trigger dsRNA molecules.

Finally, it must be mentioned that we are currently lacking a clear regulatory framework that guides and facilitates the development of this new kind of pesticide. Approaches such as plant transformation or modified insect viruses or symbionts can be quite restricted by the rigid actual regulations. Thus, the main strategy available is the exogenous application of RNA molecules. However, it is likely that we are still far from having the capacity to profitably produce the enormous quantities of RNA needed to support global agricultural production. The development of methods for overproduction of RNA has not historically accompanied that of DNA and proteins, possibly due to the difficulty related with the short half-life of RNA and the relatively minor role formerly attributed to these molecules. It was not until recently that a true revolution in exogenous dsRNA production systems began, achieving progressively higher yields at increasingly more affordable prices for large-scale use as an insecticide. Therefore, it is expected that the extensive application of RNAi strategies will come hand-in-hand with future and improved strategies based on biofactories.

## 7. Conclusions

Biotechnological applications based on RNAi may contribute to counteracting current challenges imposed by insect pests in global food production. Endogenously produced in crop plants or exogenously applied, properly engineered dsRNA molecules may substitute classic insecticides to fight insect pests in a more specific, sustainable, and environmentally friendly manner. However, for this to be true, we need to keep improving our knowledge about the insect endogenous RNAi pathways, including RNA intake and systemic movement, and to refine technologies for recombinant RNA production and delivery.

## Figures and Tables

**Figure 1 biology-13-00137-f001:**
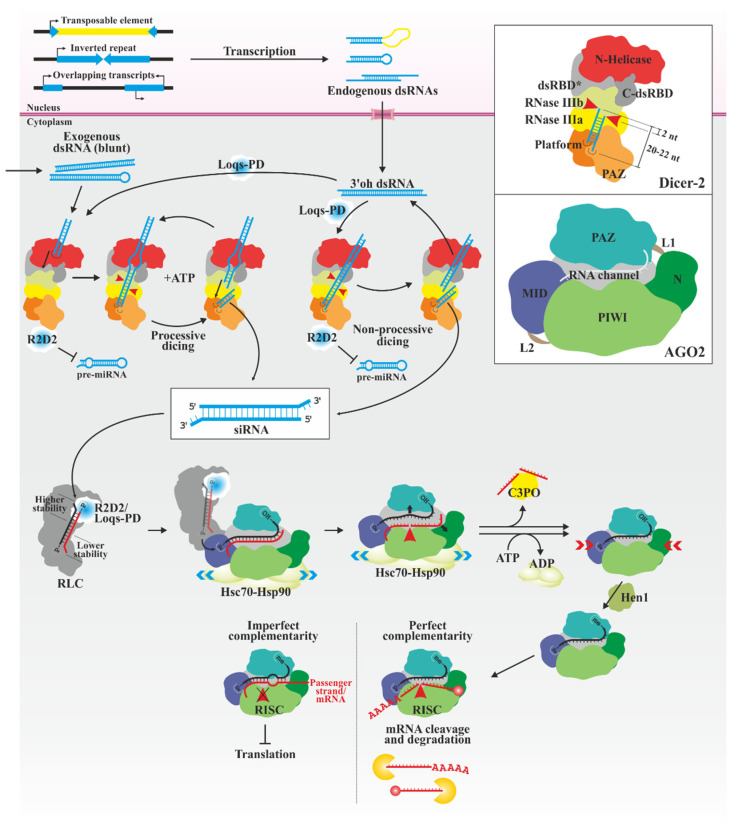
siRNA pathway in insects. Perfectly complementary long dsRNAs of exogenous or endogenous origin (depicted in blue) are differentially processed in the cytoplasm by Dicer-2 depending on the nature of their termini. Additional factors mediate the selection of suitable substrates in both processes. siRNAs of 20–22 bp are generated. Dicer-2 and R2D2 (or Loqs-PD) select the guide strand by sensing the relative stability of both ends. They transfer both chains to AGO2 in an open state, which cleaves and removes the passenger strand (red). The guide (black) is methylated at the 3′ end, and AGO2 is closed. AGO2 uses the guide strand to cleave complementary mRNAs (also in red, depicted with 5’ cap and 3’ poly(A) tail), marking them for further degradation. The presence of mismatches with the target prevents its cleavage, thus repressing mRNA translation. Inserts in the right side show the domains of Dicer-2 and AGO2 (upper and lower, respectively). The characteristic cleavage of Dicer-2 is shown in its corresponding insert. -me, -OH and -P, 3′ 2′-O-methyl, 3’-hydroxyl and 5′-phosphate termini, respectively; 3′oh, 2 nt 3′ overhang termini; AGO2, Argonaute 2; C-dsRBD, carboxyl-terminal dsRNA-binding domain; C3PO, component 3 promoter of RISC; dsRBD*, atypical dsRNA-binding domain; Hsc70-Hsp90, complex of 70 kDa heat shock analog protein and 90 kDa heat shock protein, respectively; L1 y L2, linker 1 and 2, respectively; Loqs-PD, Loquacious isoform PD; MID, middle domain; N, amino-terminal variable domain; N-helicase, amino-terminal helicase domain; PAZ, Piwi/Argonaute/Zwille domain; pre-miRNA, micro RNA precursor; R2D2, protein with two dsRNA-binding domains associated with Dicer-2; RISC, RNA-induced silencing complex; RLC, RISC loading complex; and siRNA, small interfering RNA.

**Figure 2 biology-13-00137-f002:**
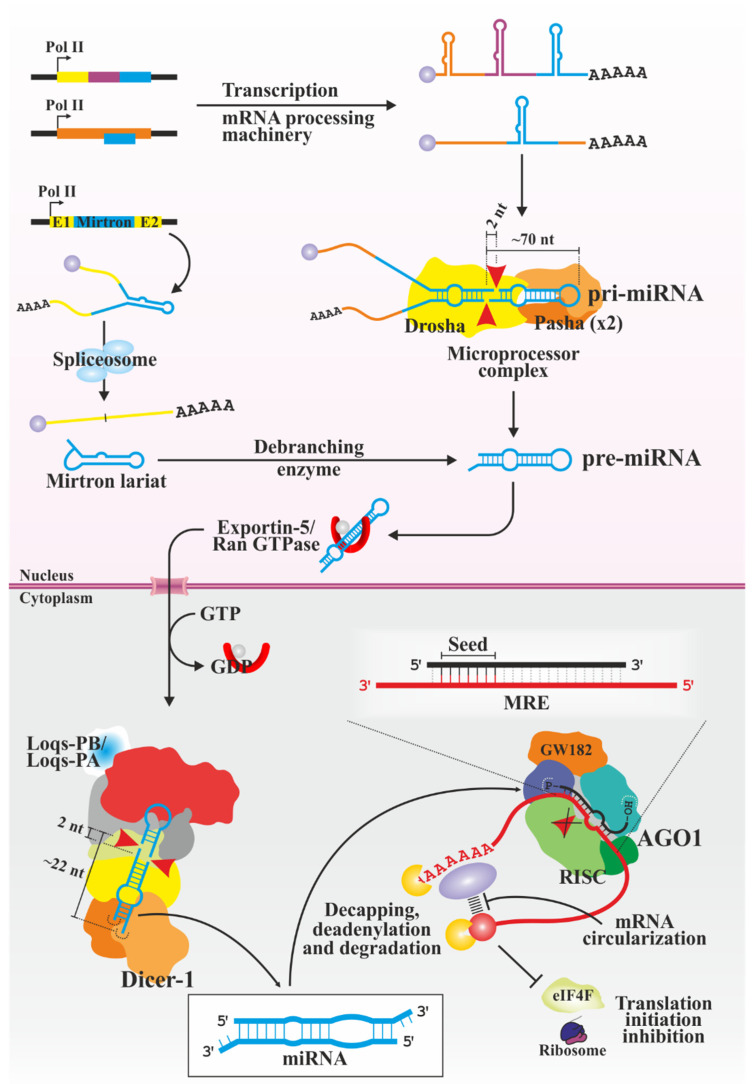
miRNA pathway in insects. miRNA-containing genomic loci are transcribed by RNA polymerase II generating long, partially dsRNA primary precursors (depicted in blue) that are trimmed into shorter hairpins by the Microprocessor complex Drosha/Pasha in the nucleus. This step is not necessary in the case of mirtrons, whose processing depends on spliceosomal and debranching machinery. The precursors are then exported into the cytoplasm where Dicer-1, aided by Loqs isoforms, eliminates the loop resulting in ~22 bp miRNA. The characteristic cleavage of the Microprocessor and Dicer-1 are shown. The miRNA guide strand (black) loaded into AGO1 induces cleavage-independent mRNA (in red, depicted with 5’ cap and 3’ poly(A) tail) degradation and translation suppression of the partially complementary mRNAs. -OH and -P, 3′-hydroxyl and 5-phosphate termini, respectively; AGO1, Argonaute 1; E1/E2, exon 1 and 2, respectively; eIF4F, eukaryotic initiation factor 4F; Loqs-PA/-PB, Loquacious isoforms PA and PB, respectively; miRNA, micro RNA; MRE, microRNA recognition elements; Pol II, RNA polymerase II; pre- and pri-miRNA, precursor and primary micro RNA, respectively; and RISC, RNA-induced silencing complex.

**Figure 3 biology-13-00137-f003:**
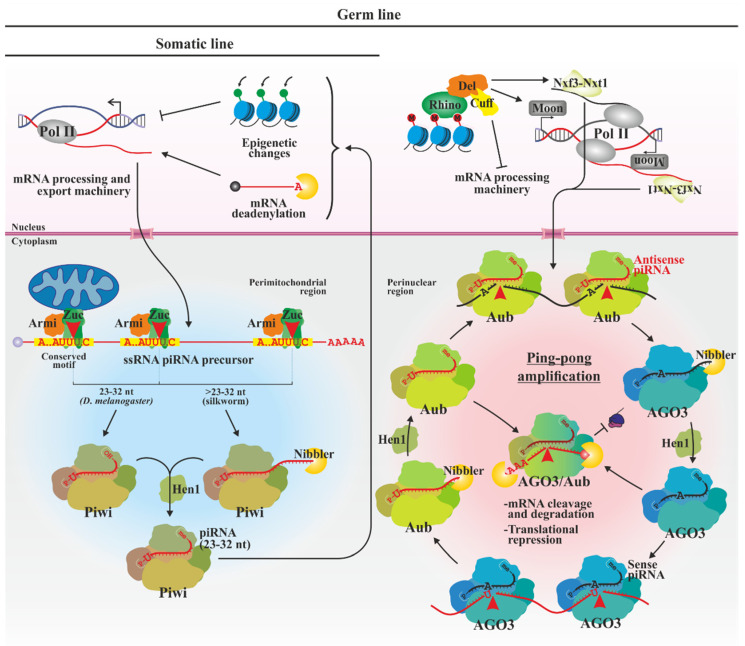
piRNA pathway in both, insects’ somatic and germline cells. In somatic cells, piRNA precursors are transcribed from unidirectional piRNA clusters. Linear precursors are exported to cytoplasmic perimitochondrial regions where Zucchini and in some cases additional exonucleases generate single-stranded 23–32 nt piRNAs that are loaded on Piwi proteins and methylated at their 3′ end. In germline, the expression of specialized transcription machinery also allows the generation of piRNAs of both senses from bidirectional piRNA clusters. Protein-loaded sense piRNAs participate in the generation of antisense piRNAs and vice versa (ping-pong cycle) in perinuclear regions. Aub/AGO3 differentially load RNAs of both polarities that are also terminally trimmed and methylated (sense in black, and antisense in red, respectively). Piwi-bound piRNAs enter the nucleus where they transcriptionally regulate gene expression while Aub/AGO3 remain in the cytoplasm and induce degradation of complementary mRNAs (also in red, depicted with 5’ cap and 3’ poly(A) tail) and repressing translation. -me, -OH and -P, 3′ 2′-O-methyl, 3’-hydroxyl and 5′-phosphate termini, respectively; AGO3, Argonaute 3; Armi, Armitage; Aub, Aubergine; Cuff, Cutoff; Del, Deadlock; M, H3K9me3 epigenetic marker; Moon, Moonshiner transcription factor; Nxf3-Nxt1, mRNA nuclear export factor complex; piRNA, PIWI-interacting RNAs; Pol II, RNA polymerase II; and Zuc, Zucchini.

**Figure 4 biology-13-00137-f004:**
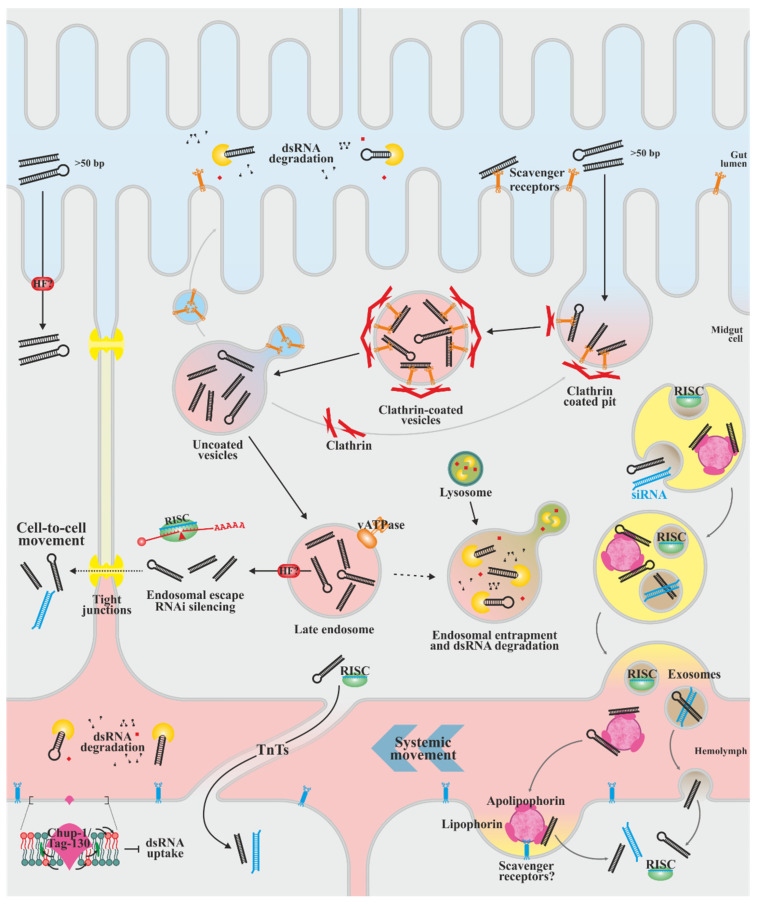
Proposed mechanisms of exogenous dsRNA cell-uptake, cell-to-cell and systemic movement in insects. Scavenger receptor-mediated clathrin-dependent endocytosis is the main pathway for cellular uptake of dsRNAs in midgut cells, although additional unknown factors may be involved in this process. The uptake is dependent on the dsRNA size, and nucleases in the digestive system may compromise the dsRNA stability. Unknown cellular factors mediate dsRNA egress from the late endosome; inefficient endosomal escape induces dsRNA degradation after endosome-lysosome fusion. In the cytoplasm, the dsRNAs mediate the silencing of endogenous host genes. dsRNAs, RNAi machinery and RNA intermediates can be transported to neighboring cells through cytoplasmic projections (TnTs) and possibly through tight junctions. Systemic movement to distant cells through hemolymph depends on exosomal encapsulation, and it has been proposed that it may also be mediated by dsRNA binding to apolipophorins. The interaction with these cellular components prevents their degradation by the hemolymph nucleases. Alterations in plasma membrane composition may be consequential in the uptake and distribution of the silencing signal. Chup-1/Tag-130, cholesterol uptake protein 1; HF?, unknown host factor; RISC, RNA-induced silencing complex; siRNA, small interfering RNA; TnTs, tunneling nanotubes; and vATPase, Vacuolar-type ATPase.

**Figure 5 biology-13-00137-f005:**
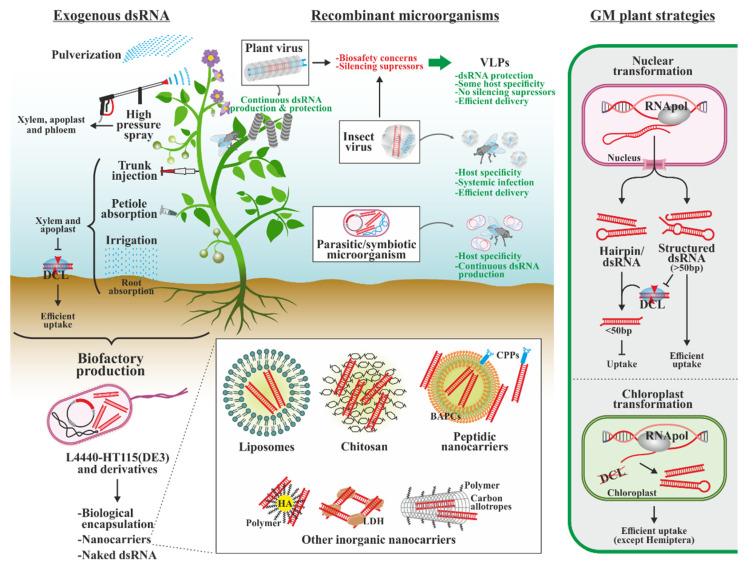
Proposed strategies for supplying dsRNAs to insects for RNAi-mediated pest control in the field. dsRNAs can be produced in biofactories and used, alone or in combination with nanomaterials (insert), with several application strategies. The use of plant and insect viruses, VLPs derived from them, as well as other insect pathogens or symbionts, taking advantage of their intrinsic characteristics, has been also studied. The development of nuclear and chloroplast transformants allows the continuous production of dsRNAs in genetically modified plants. BAPCs, branched amphipathic peptide capsules; CPPs, cell-penetrating peptides; DCL, Dicer-like proteins; GM, genetically modified; HA, hydroxyapatite; LDH, layered double hydroxide; RNApol, RNA polymerase; and VLPs, virus-like particles.

## Data Availability

Not applicable.

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
