# Peer review of "RNA Interference in Insects: From a Natural Mechanism of Gene Expression Regulation to a Biotechnological Crop Protection Promise"

_biology, 2024, doi:10.3390/biology13030137_

Round 1

Reviewer 1 Report

Comments and Suggestions for Authors

This articles reviews current knowledge about the RNAi pathways in insects, and some other applied questions such as production and delivery of recombinant RNA in insects. However, there are some concerns need to be addressed before publication.

1. As authors mentioned that lepidopteran pests have low sensitivity to RNAi. However, a considerable proportion of the pests are of the order Lepidoptera. Therefore, I hope that authors can summarize the mechanism of lepidopteran pests insensitive to RNAi, and how to design RNAi pesticides according to the characteristics of lepidopteran pests.

2. Authors should outlook the possible problems and how to overcome them in the future application of RNAi pesticides.

Comments on the Quality of English Language

Minor editing of English language required

Author Response

We thank the reviewer for his/her comments that are helping us to improve our manuscript. In the revised version, a paragraph at the end of section 5 has been added about improving RNAi strategies. A whole new section 6 about challenges and future directions have been added to the manuscript.

Reviewer 2 Report

Comments and Suggestions for Authors

These authors provide a comprehensive, compact resource article that reviews the potential of RNAi as a basis for crop insecticides. The figures provided are very detailed, the writing encompasses all topics of interest within the article's theme, appropriate sources are utilized in a broad manner to inform the article's content, and appropriate consideration for future directions is given. There are minor English language issues to be corrected, but I have no further suggested edits. I think this is a very well written manuscript.

The authors did an uncommonly good job. They covered the topic very well, and they created helpful figures. It is my informed opinion that the article does not need much more than a review for proper English language use. 325+ references is a big achievement, and this field is not that much bigger than that number. 

Comments on the Quality of English Language

Minor issues in English language use and efficiency are located throughout the article. The MDPI staff should be able to help remedy these.

Author Response

We thank the reviewer for the encouraging comments. We have scrutinized the manuscript to improve English language.

Round 2

Reviewer 1 Report

Comments and Suggestions for Authors

The author has solved almost all my concerns.

Comments on the Quality of English Language

Good.